# Shifting the Soil: Metformin Treatment Decreases the Protumorigenic Tumor Microenvironment in Epithelial Ovarian Cancer

**DOI:** 10.3390/cancers14092298

**Published:** 2022-05-05

**Authors:** Sarah E. Taylor, Daniel K. Chan, Dongli Yang, Tulia Bruno, Richard Lieberman, Javed Siddiqui, Thing Rinda Soong, Lan Coffman, Ronald J. Buckanovich

**Affiliations:** 1Division of Gynecologic Oncology, Department of Obstetrics and Gynecology, Magee-Womens Hospital of UPMC, School of Medicine, University of Pittsburgh, Pittsburgh, PA 15213, USA; dkchan21@uic.edu (D.K.C.); coffmanl@mwri.magee.edu (L.C.); buckanovichrj@mwri.magee.edu (R.J.B.); 2Magee-Womens Research Institute, Pittsburgh, PA 15213, USA; yangd6@mwri.magee.edu; 3Division of Hematology/Oncology, Department of Medicine, UPMC Hillman Cancer Center, School of Medicine, University of Pittsburgh, Pittsburgh, PA 15213, USA; 4Department of Immunology, School of Medicine, University of Pittsburgh, Pittsburgh, PA 15213, USA; tbruno@pitt.edu; 5Tumor Microenvironment Center, UPMC Hillman Cancer Center, Pittsburgh, PA 15213, USA; 6Cancer Immunology and Immunotherapy Program, UPMC Hillman Cancer Center, Pittsburgh, PA 15213, USA; 7Department of Pathology, University of Michigan, Ann Arbor, MI 48109, USA; jakrwl@med.umich.edu; 8Michigan Center for Translational Pathology, University of Michigan, Ann Arbor, MI 48109, USA; siddiqui@med.umich.edu; 9Department of Pathology, School of Medicine, University of Pittsburgh, Pittsburgh, PA 15213, USA; soongt@upmc.edu

**Keywords:** epithelial ovarian cancer, tumor microenvironment, cancer-associated mesenchymal stem cells, metformin

## Abstract

**Simple Summary:**

Metformin is a drug commonly used to treat diabetes but it may play a role in cancer treatment. It is not known exactly how metformin acts on cancer cells and cells in the tumor microenvironment (TME). Our previous work suggested that metformin may be altering specific cells in the tumor microenvironment called cancer-associated mesenchymal stem cells (CA-MSC). The aim of this study was to build on our previous work to understand the impact of metformin on the TME. We demonstrated that when individuals with epithelial ovarian cancer received metformin in addition to chemotherapy, their tumors were less likely to have CA-MSC and T regulatory cells, which are both known to promote tumor growth, when compared to tumors from individuals who received chemotherapy alone. Additional experiments with ovarian cancer cells and tumors grown in mice suggested that metformin might be best used to prevent tumor growth rather than treat advanced stage disease.

**Abstract:**

Controversy persists regarding metformin’s role in cancer therapy. Our recent work suggested metformin acts by impacting the tumor microenvironment (TME), normalizing the epigenetic profile of cancer-associated mesenchymal stem cells (CA-MSC). As CA-MSC can negatively impact tumor immune infiltrates, we evaluated metformin’s impact on the human TME, focusing on the interplay of stroma and immune infiltrates. Tumor samples from (i) 38 patients treated with metformin and chemotherapy and (ii) 44 non-metformin matched controls were included in a tissue microarray (TMA). The TMA was used to compare the presence of CA-MSC, desmoplasia and immune infiltrates in the TME. In vitro and in vivo models examined metformin’s role in alteration of the CA-MSC phenotype. The average percentage of CA-MSC was significantly lower in metformin-treated than in chemotherapy alone-treated tumors (*p* = 0.006). There were fewer regulatory T-cells in metformin-treated tumors (*p* = 0.043). Consistent with CA-MSC’s role in excluding T-cells from tumor islets, the T-cells were primarily present within the tumor stroma. Evaluation of metformin’s impact in vitro suggested that metformin cannot reverse a CA-MSC phenotype; however, the in vivo model where metformin was introduced prior to the establishment of the CA-MSC phenotype supported that metformin can partially prevent the reprogramming of normal MSC into CA-MSC. Metformin treatment led to a decrease in both the presence of protumorigenic CA-MSC and in immune exclusion of T cells, leading to a more immune-permissive environment. This suggests clinical utility in prevention and in treatment for early-stage disease and putatively in immune therapy.

## 1. Introduction

Epithelial ovarian cancer (EOC) has the third highest mortality:incidence ratio of all cancers, making it the deadliest of all gynecologic cancers and fifth leading cause of cancer death in women [1]. While most patients with EOC have tumors that respond to frontline chemotherapy, approximately 75% of patients who initially respond to therapy will ultimately experience cancer relapse and succumb to their disease [2]. Research efforts have focused on transforming the initial chemosensitivity into a durable response. To this end, researchers have investigated the addition of anti-angiogenic drugs, such as bevacizumab, which has repeatedly demonstrated an incremental improvement in progression-free survival with overall survival benefit limited to a “high risk” group [3,4]. More recently, there have been promising results with the addition of poly ADP-ribose polymerase (PARP) inhibitors with documented improvement in overall survival in women with *BRCA* pathogenic variants who received a PARP inhibitor as maintenance therapy after treatment for platinum-sensitive recurrent disease [5]. Unfortunately, neither of the approaches has yet to lead to the long sought-after initial durable response in the broader population of patients with EOC. Moreover, these therapies come with a high financial burden [6,7]. Thus, the development of novel, affordable therapeutic approaches is needed. 

Metformin represents one potential simple, well-tolerated, new therapeutic approach. Metformin belongs to a class of drugs called biguanides and is the first-line pharmacologic therapy for individuals with type II diabetes mellitus (T2DM) because of its glycemic efficacy, lack of weight gain and hypoglycemia, overall tolerability, and low cost [8]. Epidemiologic data support an anti-cancer action of metformin across cancer types, including EOC [9,10,11,12,13,14,15]. These studies indicated that compared with diabetics on other treatments, diabetics on metformin have a lower risk of ovarian cancer [13,14] and a lower ovarian cancer mortality [15]. Preclinical studies demonstrated that metformin has anticancer activity, but the exact mechanism has yet to be elucidated [16,17,18,19,20,21,22,23]. Based upon this information, Brown et al. ran a single arm phase II clinical trial of metformin with chemotherapy as adjuvant therapy for newly diagnosed EOC with a resultant median progression-free survival (PFS) of 18.0 months and median overall survival (OS) of 57.9 months [24]. While the PFS is similar to previous reports, the OS represents a significant improvement.

How metformin impacts outcomes of cancer patients remains unknown and is a topic of debate. Multiple areas of study have been undertaken to elucidate the effects of metformin on the tumor microenvironment (TME). One new target of interest in the TME is mesenchymal stem cells (MSCs). MSCs are non-hematopoietic multipotent stem cells that can differentiate to create numerous protumorigenic components of the TME including fibroblasts and adipocytes [25]. MSCs within the ovarian TME are epigenetically reprogrammed to have a distinct expression profile relative to normal MSCs [24,26]. These ovarian carcinoma-associated MSCs (CA-MSCs) contribute to a pro-tumorigenic TME, increasing cancer stemness and driving chemotherapy resistance and metastasis [26,27,28,29,30,31,32,33,34]. Furthermore, recent studies demonstrated that CA-MSCs are highly immunosuppressive and drive tumor immune exclusion, a strong marker of poor patient prognosis [35,36].

A potential therapeutic role for metformin impacting tumor-mediated CA-MSC reprogramming was recently implicated. Translational work done in conjunction with the clinical trial of metformin demonstrated significant changes in the DNA methylation of CA-MSCs [24]. Analysis of CA-MSC DNA methylation from metformin-treated trial patients found that in 6 of 11 patients, CA-MSCs looked more like normal MSCs than CA-MSCs from control patients. The remaining five clustered with the control CA-MSCs. The patients whose CA-MSCs were not modified by metformin treatment had a poor outcome compared to patients whose CA-MSCs showed a normalization with therapy [24].

Given that the findings related to CA-MSC alteration in the aforementioned study had a limited number of patients, in this study we sought to evaluate the relationship between CA-MSC and metformin in a larger number of patients. Additionally, since metformin and CA-MSC have been linked with alterations in immune cells, we evaluated the impact of metformin on immune cell infiltrates in the ovarian TME. Consequently, we present an expanded analysis of the impact of metformin treatment on the tumor microenvironment, including the interplay of the stroma, the immune infiltrates and the tumor, and its role in the treatment effect of patients with ovarian cancer who were treated with metformin as part of a previously reported phase II clinical trial [24].

## 2. Materials and Methods

### 2.1. Tissue Samples

Metformin-treated tumors were collected from a previously reported single-center, open label phase II clinical trial of patients with a new diagnosis of confirmed advanced-stage epithelial ovarian cancer without a diagnosis of diabetes mellitus or metformin use in the preceding six months [24]. All patients gave written informed consent before participation in the study. Patient samples were obtained in accordance with protocols approved by the University of Michigan Institutional Review Board (IRB). A tissue microarray (TMA) was constructed from tissue samples from patients that were enrolled in the trial and had an available tumor as well as tissue from patients with a new diagnosis of epithelial ovarian cancer that received chemotherapy without metformin but were matched based on treatment, patient age, and tumor stage [24]. Each individual tumor included in the TMA had samples in triplicate. Histology was confirmed and cores evaluated independently by two gynecologic pathologists (R.L. and T.R.S.) to have at least a 50% epithelial component.

### 2.2. Desmoplasia Calculation

Utilizing a hematoxylin and eosin (H&E) slides made from the TMA, each individual core was marked with areas of desmoplasia. Desmoplasia was defined as the regions between cytokeratin-positive tumor islets. Areas at least 20 µm wide were evaluated. Individual areas were then totaled and divided by the area of the total core. Cores with low cellularity were excluded. What remained of the triplicates was then averaged across the cores to give a single mean value for each individual patient.

### 2.3. Multispectral Spatial Analysis

The TMA was stained with immunofluorescent markers including CD4, Foxp3, CD8, CD20, CD68, panCK, and DAPI as previously described [35,37]. The TMA was then analyzed using a Vectra Automated Quantitative Pathology Imaging System (Akoya Biosciences, Marlborough, MA, USA), which accurately detects and measures weakly expressed and overlapping biomarkers via immunofluorescence. Tissue segmentation on the TMA was defined as “tumor” for those cells that were panCK/DAPI-positive and “other” for stroma or cellular tissue without panCK/DAPI expression. Phenotypes counted included CD4^+^/Foxp3^−^ helper T cells (Th-cells), CD4^+^/Foxp3^+^ T regulatory cells (T-regs), CD8^+^ cytotoxic T cells (Tc-cells), CD20^+^ (B-cells), CD68^+^ (macrophages), and CK^+^ (tumor cells). 

Cell counts were reported for each phenotype across each segmentation for each individual core. Cores with tissue damage in processing were excluded. What remained of the triplicates was then averaged across the cores to give a single value for each individual patient.

To quantify the presence and amount of carcinoma-associated mesenchymal stem cells, utilizing the International Society for Cellular Therapy (ISCT) criteria for defining multipotent mesenchymal stromal cells [38], serial sections from the TMA were independently stained with CD73 (BD Pharmingen, San Diego, CA, USA; red), CD90 (BD Pharmingen, green), and WT-1 (R&D Systems, Minneapolis, MN, USA; blue) as previously described [35]. Individual images for each stain were captured for each individual sample, and then overlayed and analyzed using Image J [39] to quantify the percentage of CA-MSCs present. 

### 2.4. Coculture CA-MSC Classification

Omental-derived normal MSCs or CA-MSCs were derived from surgical resections of the human omentum obtained at the time of surgery for benign indication or for ovarian, fallopian, or primary peritoneal cancer, respectively. Patient samples were obtained in accordance with protocols approved by the University of Pittsburgh IRB. MSC or CA-MSC were grown in vitro with or without CAOV3 coculture. CAOV3 cells were purchased from ATCC. Cells were cultured in a 1:1 media of RPMI-1640 medium, with 10% FBS and 1% penicillin/streptomycin, MEBM media at 37 °C and 5% CO_2_ as previously described [26]. In vitro cultures were exposed to metformin (100 µM, 1 mM) or vehicle (sterile water) for 7 d and then RNA was harvested (RNeasy, Qiagen, Germantown, MD, USA) and qRT-PCR performed to generate a CA-MSC probability score using the previously described ovarian CA-MSC classifier [26]. 

Briefly, a CA-MSC classifier score was developed based on RNA sequencing studies that revealed significant expression differences between normal MSCs (derived from normal omentum) and CA-MSCs (derived from ovarian cancer omental metastatic deposits). Differential gene expression enrichment (using the Sergushichev fast algorithm, at a <0.05 adjusted *p* value level) identified 27 genes that discriminated between CA-MSCs and omental MSCs. Following independent cross-validation, a constrained (LASSO) logistic regression modeling was applied to identify a PCR expression-based classifier that predicts the probability of being a CA-MSC versus MSC. The sum of regression coefficients*gene PCR expression was logistically transformed yielding a score between 0 and 1. Scores closest to 1 had the highest probability of being a CA-MSC and scores closest to 0 were most likely to be a normal MSC. The final model incorporated expression data from six genes: Annexin A8-like protein 2 (ANXA8L2), Collagen Type XV Alpha 1 Chain (COL15A1), Cytokine Receptor Like Factor 1 (CRLF1), GATA Binding Protein 4 (GATA4), Iroquois Homeobox 2 (IRX2), and TGF-β2 [26].

### 2.5. In Vivo Mouse MSC/CA-MSC Tumor Xenograft Model

All studies were performed with the approval of the University of Pittsburgh IRB. CAOV3 and normal omental MSC, or CA-MSC were independently cultured and then co-injected into the flank of NSG mice in a 1:1 concentration as previously described [33]. A total of 72 h after tumor injection, mice were treated daily with metformin 150 mg/kg—a dose that achieves blood levels similar to those seen in a patient receiving ~750 mg daily [40]. Tumor growth was monitored, and tumors were resected when they reached ~400 mm^3^. Tumors were then dissociated into live single cell suspensions and CD73^+^, CD90^+^, and CD105^+^ MSC FACS were isolated and then expanded as previously described [26]. If sufficient numbers of MSCs were not isolated, cells were plated in MEBM media to expand MSCs and were then FACS-sorted as above. RNA was harvested and qRT-PCR was performed to generate a CA-MSC probability score as described above [26]. 

### 2.6. Statistical Analysis

Significance was determined at a significance threshold of *p*-values (*p* < 0.05) unless otherwise specified. Descriptive data were summarized with percentages, means, and standard deviations. A two-sample Wilcoxon rank-sum (Mann–Whitney) test was used to compare the immunophenotypes, desmoplasia, and quantity of CA-MSCs for tumors treated with metformin compared to those that were not. All in vitro experiments were performed in triplicate samples and repeated at least three times, unless specified otherwise.

## 3. Results

### 3.1. Clinical Characteristics

Tumor samples from the 38 evaluable patients included in the phase II clinical trial of metformin plus platinum-based standard of care chemotherapy were included in the construction of the tissue microarray (TMA) [24]. Tumor samples from 44 non-metformin-treated control patients, matched for age (within 5 years), stage, and treatment, treated at the same institution in a similar timeframe (2012–2017) were included as control samples in the TMA. Clinical characteristics and outcomes from the trial have been previously published [24]. In brief, the cohort included patients with predominantly stage III disease (65.8%) and with a majority high grade serous histology (76.3%). Similar to the group treated with metformin, the control group mostly included patients with stage III disease (68.2%) and again predominantly those with high-grade serous histology (65.9%). 

### 3.2. Metformin Decreased the Presence of Carcinoma-Associated Mesenchymal Stem Cells (CA-MSCs)

We have previously shown that in a subset of metformin-treated patients, CA-MSCs demonstrated a ‘normalization’ of their DNA methylation profile. Tumors from patients that had CA-MSC normalization had a reduction in cancer stemness and chemotherapy resistance ex vivo [24]. Furthermore, normalized CA-MSCs lost the ability to promote chemotherapy resistance ex vivo. To expand the evaluation of CA-MSCs in metformin-treated tumors and controls we performed multispectral fluorescent analysis for CA-MSC markers to quantify the presence of CA-MSCs in tumors from patients treated with metformin (*n* = 29) compared to those treated with chemotherapy alone (*n* = 33). Human CA-MSCs express CD90 and CD73. CA-MSCs can be distinguished from normal MSCs based on the expression of WT1 [34]. Using multispectral analysis for these three markers, we quantified the number of CA-MSCs in control and metformin-treated tumors. As anticipated, CA-MSCs were primarily found in the peritumoral stroma (Figure 1A). The average percentage of WT1^+^, CD90^+^, CD73^+^ CA-MSC in tumors exposed to metformin (0.19%, SD 0.23%) was significantly lower than in tumors treated with standard chemotherapy alone (0.52%, SD 0.89%) (*p* = 0.006) (Figure 1B). 

### 3.3. Metformin Did Not Impact Desmoplasia

As it has previously been demonstrated that MSCs impact desmoplasia and desmoplasia can lead to tumor immune exclusion and worse clinical outcomes [25,41,42], we sought to quantify the amount of desmoplasia present in the tumor samples in the TMA and compare those treated with or without metformin. While the tumors treated with metformin (*n* = 29) did have a lower average percentage of desmoplastic tissue (34.9%, SD 19.9%), this was not statistically significantly different from those treated with chemotherapy alone (*n* = 33) (37.5%, SD 16.2%) (*p* = 0.412) (Figure 2).

### 3.4. Metformin Treatment Was Associated with a Reduction in Regulatory T Cells

CA-MSCs play a critical role in the recruitment of immune-suppressive myeloid cells [43], which can result in the induction of regulatory T cells [44] and tumor-immune exclusion of immune-effector cells [25,35,45]. We therefore looked at the presence and composition of the immune cells in tumor islet and in the surrounding stroma. Specifically, we analyzed CD4^+^/Foxp3^+^ T regulatory cells (T-regs), CD4^+^/Foxp3^−^ helper T cells (Th-cells), CD8^+^ cytotoxic T cells (Tc-cells), CD20^+^ B-cells, CD68^+^ macrophages, and CK^+^ tumor cells (Figure 3A).

Analysis of T cells indicated that there were significantly more total regulatory T-cells (T-regs -CD4^+^/Foxp3^+^) (*p* = 0.043) in control tumors compared to metformin-treated tumors. These T-regs were primarily found in the tumor stroma (control vs. metformin-treated, *p* = 0.019). Similarly, metformin treatment was associated with a reduction in the number of CD4^+^/Foxp3^−^ T helper cells (Th-cells, *p* = 0.002) with the primary driver being the difference in the number of CD4^+^/Foxp3^−^ Th-cells within the stroma (*p* = 0.003). While the number of intra-islet T-cells was relatively low compared to the number of T-cells in the stroma, there were similarly more CD4^+^/Foxp3^−^ Th-cells in the tumor islet in the control samples (*p* = 0.043). Analysis of CD8^+^ cytotoxic T cells (Tc-cells) demonstrated that, like the CD4^+^ T helper cells, there was an increase in the total number of CD8^+^ Tc-cells in control tumors, with these Tc-cells primarily restricted to the tumor stroma (*p* = 0.017). There was no difference in the number of CD8^+^ Tc-cells found in tumor islets (*p* = 0.077). Analysis of CD68^+^ macrophages indicated a trend for a reduction in macrophages in metformin-treated tumors, but no statistical difference. Similarly, CD20^+^ B-cells showed no difference between control or metformin-treated tumors neither in the stroma (*p* = 0.130) nor in the tumor islet (*p* = 0.767) (Figure 3B).

### 3.5. Metformin Could Not Reverse CA-MSC Phenotype In Vitro

We previously showed that CA-MSCs are derived from local tissue MSCs that are epigenetically reprogrammed by the TME [24]. As such, the metformin-related reduction in CA-MSCs observed above could be due to (i) a reversion of the CA-MSC phenotype back to a normal MSC, or (ii) a prevention of the induction of the CA-MSC phenotype. To determine if metformin could reverse the CA-MSC phenotype, we first treated CA-MSCs derived from three different chemotherapy-naïve patients with metformin in vitro for one week and then evaluated their profile using a previously developed mRNA expression-based CA-MSC classifier score [26]. A classifier score higher than 0.95 indicates a CA-MSC phenotype, while a score less than 0.3 is indicates a normal MSC. In vitro treatment of these established CA-MSCs had no impact on the classifier score (0.99 and 1.00, Figure 4A). This suggests that metformin, at least in vitro, cannot reverse the CA-MSC phenotype. 

### 3.6. Metformin Has the Potential to Prevent the Reprogramming of MSC to a CA-MSC Phenotype

To determine if metformin can prevent the induction of a CA-MSC phenotype, we used a previously developed in vivo tumor model. We have previously demonstrated that normal tissue-derived MSCs (either omentum or ovarian tissue derived), when grown together with tumor cells in vivo, undergo epigenetic reprogramming to a CA-MSC phenotype [24]. To determine if metformin could prevent the induction of a CA-MSC phenotype generated by the MSC/CAOV3 tumor xenografts, we initiated therapy with metformin or vehicle control three days after tumor injection. As a control, CA-MSCs were similarly engrafted with CAOV3. When tumors reached ~400 mm^3^, animals were euthanized and MSCs were FACS were isolated from tumors. MSCs were then analyzed using the CA-MSC classifier to determine if they had been reprogrammed to CA-MSCs. MSCs isolated from tumors established with CA-MSC had a CA-MSC classifier score of 0.99, indicating a CA-MSC phenotype. Similarly, as expected, human MSCs isolated from tumors established with normal MSCs and treated with a vehicle control had a classifier score consistent with reprogramming to CA-MSCs (classifier scores of 0.98–0.99 among three replicates representing three individual tumors from three independent mice). By contrast, human MSCs isolated from tumors established with normal MSCs and treated with metformin had a classifier score (0.3 and 0.7) that was not consistent with CA-MSCs (Figure 4B). This indicates that metformin can at least partially prevent the reprogramming of MSCs to a CA-MSC phenotype. 

## 4. Discussion

While there have been some advances in the treatment of epithelial ovarian cancer, most patients will ultimately succumb to their disease. There is a clear, ongoing need to identify therapeutic interventions to improve outcomes. Metformin represents one potential simple, cheap, generally well-tolerated, and new therapeutic approach. While there is an abundance of epidemiologic data touting the potential benefits, the anti-cancer mechanism of action has yet to be determined. 

Building on our previous work, we demonstrate that metformin is impacting multiple facets of the tumor microenvironment. Consistent with our previous findings that metformin could impact the methylation pattern of CA-MSCs [24], we find that metformin treatment is associated with a decreased number of CA-MSCs present in the tumor. This was specifically linked to WT1 expression, as WT1 was recently linked as a critical driver of CA-MSCs [34]. Despite the reduction in CA-MSCs, there was no statistical difference in the amount of desmoplasia noted between the two cohorts. This may be a technical artifact related to the construction of the TMA, as tissues were selected to have a pre-specified cut-off of at least 50% presence of tumor cells. In addition, patients were treated with only 7–10 days of metformin prior to tumor resection, and this short duration of metformin use could limit metformin’s effect. Finally, normal MSCs (non-reprogrammed) in the TME can also contribute to desmoplasia. 

Associated with the decrease in CA-MSCs, we observed a significant difference in the amount of specific immune infiltrates in the tumor stromal compartment of the tumor microenvironment. Specifically, we observed that compared with controls there were significantly fewer CD4^+^ T-regs and T-helper cells in the metformin-treated tumors. This is consistent with recent observations that CA-MSC can drive an immune-suppressive TME [35]. This would also be in line with reports that metformin can increase the activity of immune checkpoint inhibitor therapy [46]. Prior work suggests that MSCs act by increasing the recruitment of M2-like macrophages [44]. While there was a trend for a reduction of TAMs in the metformin-treated tumors, this was not statistically significant. 

Potentially explaining the unique phenotype observed in the metformin-treated tumors, the in vivo model where metformin was introduced prior to the establishment of the CA-MSC phenotype supports that metformin can prevent the epigenetic reprogramming of normal MSCs into CA-MSCs. This is consistent with our prior work, which noted that a proportion of metformin-treated patients had tumors in which MSCs had a normal or intermediate phenotype and did not have the CA-MSC phenotype [24]. These results potentially explain clinical trial findings where metformin was relatively ineffective as a therapy in patients with advanced cancer [47]. Indeed, in our clinical trial, patients with stage II or III disease appeared to have a clinical benefit compared to historical controls, whereas patients with stage IV disease did not. The studies performed here offer important clinical insights on the appropriate use of metformin in cancer care, suggesting the greatest clinical benefit may be in prevention studies and in patients with early-stage disease.

## 5. Conclusions

In summary, we find that metformin treatment is associated with a decrease in the presence of protumorigenic CA-MSCs in epithelial ovarian cancer. This is associated with a decrease in immune exclusion of T-cells, suggesting that metformin, via reduction in CA-MSCs, could be leading to a more immune-permissive environment. This is consistent with the lack of impact on progression-free survival with a prolonged overall survival associated with metformin treatment. Furthermore, we find that metformin may act by preventing the reprogramming of local tissue MSCs into CA-MSCs. This suggests that metformin can be best used clinically as a preventative therapy and in patients with early-stage disease. 

## Figures and Tables

**Figure 1 cancers-14-02298-f001:**
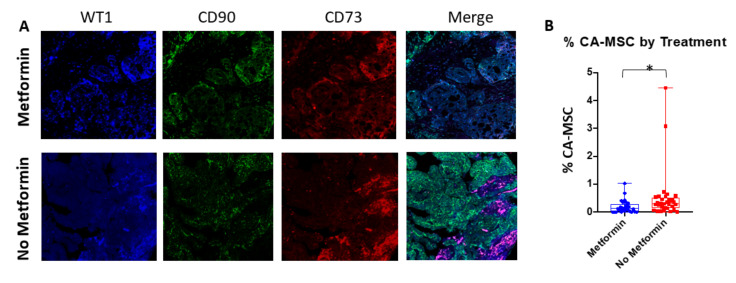
Treatment with metformin is associated with decreased number of CA-MSC. (**A**) Representative confocal microscopy immunofluorescent images of metformin-treated and non-metformin-treated human samples (10×). Overlays of CD73 (red), CD90 (green), and WT1 (blue) identify CA-MSCs (magenta). (**B**) Quantification of CA-MSCs for available tissue samples. CA-MSCs reported as a percentage of the total tissue area. * indicates statistical significance at *p* < 0.01.

**Figure 2 cancers-14-02298-f002:**
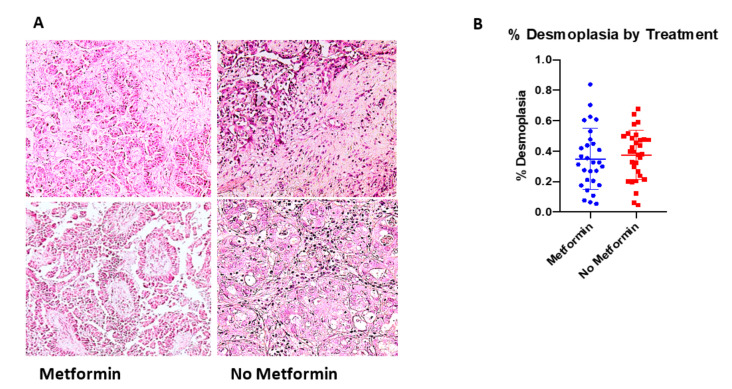
Amount of desmoplasia did not differ with addition of metformin to standard chemotherapy (control). (**A**) Representative light microscopy images of H&E stained tumor tissue from metformin-treated and non-metformin-treated human samples (10×). (**B**) Quantification of desmoplasia amount for available tissue samples. Desmoplasia reported as a percentage of the total tissue area.

**Figure 3 cancers-14-02298-f003:**
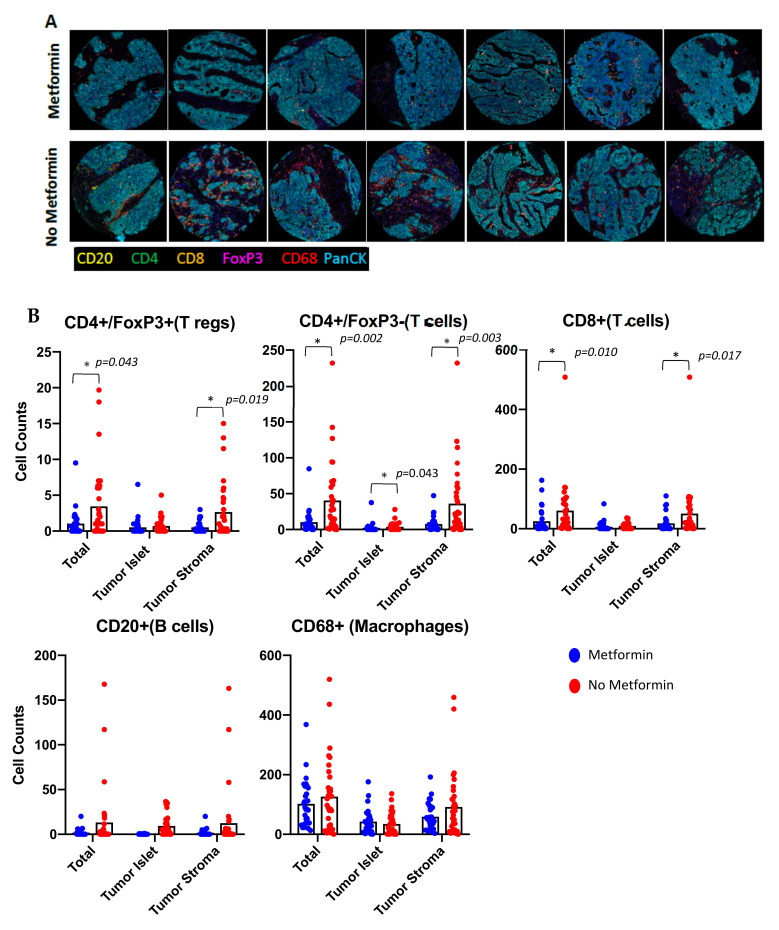
Metformin therapy was associated with a reduction in the number of T-regs. (**A**) Representative confocal microscopy immunofluorescent images of metformin-treated and non-metformin-treated human samples (10×). (**B**) Quantification of immunologic phenotypes across tumor islet and tumor stromal segments as well as total cores. Phenotypes counted included CD4^+^/Foxp3^−^ (Th-cells), CD4^+^/Foxp3^+^ (T-regs), CD8^+^ (Tc-cells), CD20^+^ (B-cells), CD68^+^ (Macrophages), and CK^+^ (Tumor Cells). * indicates statistical significance.

**Figure 4 cancers-14-02298-f004:**
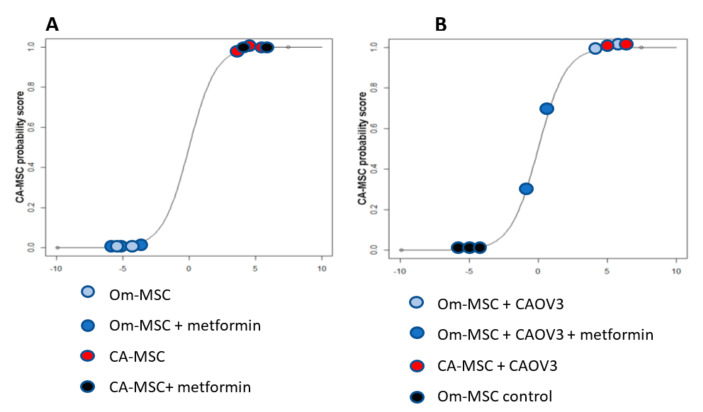
Metformin can prevent the conversion of MSCs to CA-MSCs. (**A**) CA-MSCs derived from chemotherapy-naïve patients did not show a change in the CA-MSC classifier score (0.99–1.00) when treated with metformin in vitro for one week. (**B**) MSC classifier score from MSCs and CA-MSCs isolated from tumor cells:MSC xenografts treated with metformin. MSCs isolated from tumors established with CA-MSCs had a CA-MSC phenotype (0.99). MSCs isolated from tumors established with normal omental MSCs and treated with a vehicle control had a classifier score consistent with reprogramming to CA-MSCs (classifier scores of 0.98–0.99). Human MSCs isolated from tumors established with normal MSCs and treated with metformin had a classifier score (0.3 and 0.7) that was not consistent with CA-MSCs.

## Data Availability

This data have previously been reported in [23].

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
