# Peer review of "Shifting the Soil: Metformin Treatment Decreases the Protumorigenic Tumor Microenvironment in Epithelial Ovarian Cancer"

_cancers, 2022, doi:10.3390/cancers14092298_

Round 1

Reviewer 1 Report

The manuscript "Shifting the Soil: Metformin Treatment Decreases the Protumorigenic Tumor Microenvironment in Epithelial Ovarian Cancer" is well written and the experiments are well describe. Conclusion are coherent and a logical consequence of the experiments performed.

The problem with metformin is that is still unresolved the exact mechanism of action. The present work identify a possible role for the drug, open the way for further investigation in the precise molecular mechanism of metformin action.

Author Response

Thank you for your supportive comments.

Reviewer 2 Report

Authors describe the impact of metformin treatment on epithelial ovarian cancer patients, in particular showing a reduction of cancer-associated MSC and T regulatory cells. This is an interesting observation which might be translated in the clinic. However, reduction in CA-MSC and Treg is not supported by clear in vitro tests. Further, metformin does not affect desmoplasia and cannot reverse CA-MSC phenotype, but only partially prevent the epigenetic reprogramming of MSC to CA-MSC. Thus, many other in vitro evidence is necessary to suggest metformin in ovarian cancer, although as a preventive therapy. It is necessary to show using in vitro/in vivo analysis the real impact of metformin in reducing CA-MSC and Treg enrichment in tumor mass.

Author Response

Point 1: Authors describe the impact of metformin treatment on epithelial ovarian cancer patients, in particular showing a reduction of cancer-associated MSC and T regulatory cells. This is an interesting observation which might be translated in the clinic. However, reduction in CA-MSC and Treg is not supported by clear in vitro tests. Further, metformin does not affect desmoplasia and cannot reverse CA-MSC phenotype, but only partially prevent the epigenetic reprogramming of MSC to CA-MSC. Thus, many other in vitro evidence is necessary to suggest metformin in ovarian cancer, although as a preventive therapy. It is necessary to show using in vitro/in vivo analysis the real impact of metformin in reducing CA-MSC and Treg enrichment in tumor mass.

Response 1: Thank you for your thoughtful comments.  This work sought to analyze human samples which precludes true in vitro and in vivo analyses.  Data does exist examining the impact of metformin on immune infiltrates in mouse models (Scharping NE, Menk AV, Whetstone RD, Zeng X, Delgoffe GM. Efficacy of PD-1 Blockade Is Potentiated by Metformin-Induced Reduction of Tumor Hypoxia. Cancer Immunol Res. 2017 Jan;5(1):9-16. doi: 10.1158/2326-6066.CIR-16-0103. Epub 2016 Dec 9. PMID: 27941003; PMCID: PMC5340074).  However, our goal was to examine the impact on human samples.

We would like to point out that both in vivo and in vitro murine studies were completed to evaluate the impact of metformin on CA-MSC generation.  However, as we are studying human MSC, it necessitates that studies be done in immune-suppressed mice and evaluation of the impact on immune cells in not possible.

The fact that changes were not noted in desmoplasia has several possible explanations that make interpretation of the impact of metformin on desmoplasia challenging. First, human samples that were chosen specifically for the construction of a TMA had a pre-specified cut off of at least 50% presence of tumor cells.  Thus, the lack of difference in the amount of desmoplasia noted between the two treatment groups could represent bias based on core selection. Further, patients were treated with only 7-10 days of metformin and this short duration of metformin use could limit metformin effect. Finally, normal MSCs could be contributing to desmoplasia. Because there is not a clear explanation, this is included as part of the discussion, lines 312-324.

We agree that metformin may be best utilized as a preventative measure.  Teasing out that role in mouse models are challenging and would require significant additional time and resources and are beyond the scope of the current work.  As such, these additional studies were not included in this manuscript.

Reviewer 3 Report

The manuscript is clear and very interesting; the authors demonstrate that metformin administration in patients with ovarian cancer reduces the presence of pro-tumorigenic CA-MSC and leads to a more immune permissive environment. The only problem I find is related to reference to epigenetic reprogramming. Indeed in paragraph 3.6 (line 274), the authors  say that metformin has the potential to prevent the epigenetic reprogramming of MSC to a CA-MSC phenotype, but in the manuscript there are no experiments studying epigenetics. Moreover, authors cite the reference 26, lines 277-279: “We have demonstrated that normal  tissue derived MSC ..when grown together with tumor cells in vivo undergo epigenetic reprogramming to a CA-MSC phenotype”, but the paper ref. 26 does not mention any epigenetic study (was it ref.24??). Also, the abstract (lines 44-45) “Evaluation of metformin’s impact in vitro suggested that metformin …can at least partially prevent  the epigenetic reprogramming of MSC to a CA-MSC phenotype in vivo” should be modified, while the sentence in the discussion (line 335) can be kept unchanged.

Minor points:

- in Figure 1A “Metformin” and “no metformin” are cut.

-in figure 1B CAMSC should be changed to CA-MSC.

Author Response

Point 1: The manuscript is clear and very interesting; the authors demonstrate that metformin administration in patients with ovarian cancer reduces the presence of pro-tumorigenic CA-MSC and leads to a more immune permissive environment. The only problem I find is related to reference to epigenetic reprogramming. Indeed in paragraph 3.6 (line 274), the authors  say that metformin has the potential to prevent the epigenetic reprogramming of MSC to a CA-MSC phenotype, but in the manuscript there are no experiments studying epigenetics. Moreover, authors cite the reference 26, lines 277-279: “We have demonstrated that normal  tissue derived MSC ..when grown together with tumor cells in vivo undergo epigenetic reprogramming to a CA-MSC phenotype”, but the paper ref. 26 does not mention any epigenetic study (was it ref.24??). Also, the abstract (lines 44-45) “Evaluation of metformin’s impact in vitro suggested that metformin …can at least partially prevent  the epigenetic reprogramming of MSC to a CA-MSC phenotype in vivo” should be modified, while the sentence in the discussion (line 335) can be kept unchanged.

Response 1: Thank you for your thorough review and insights. We apologize for the mistake related to our references. We have updated the references for lines 277-279 from 26 to 24 as 24 included the epigenetic evaluation that was referenced in the text and clarified that this was demonstrated in previous work.   Furthermore, the reviewer is correct in that we have not explicitly shown epigenetic reprogramming in this study.  We have modified the text as suggested to remove statements related to epigenetic reprogram except as related to prior work.

The line in the abstract (lines 44-45) has been updated to reflect the information that “metformin can at least partially prevent MSC reprograming into CA-MSC”.

Point 2: in Figure 1A “Metformin” and “no metformin” are cut.

Response 2: Figure 1A has been updated so the labels are now clearly visible.

Point 3: in figure 1B CAMSC should be changed to CA-MSC.

Response 3: Figure 1B has been updated to the include the hyphen as suggested.

Round 2

Reviewer 2 Report

I'm sorry but without any useful additional in vitro experiments to the
data presented, it is my opinion that
the work is unable to give a clear message to a reader.

Author Response

We appreciate the reviewers comments.  As mentioned previously, this work sought to analyze human samples which precludes true in vitro and in vivo analyses.

We would like to point out that both in vivo and in vitro murine studies were completed to evaluate the impact of metformin on CA-MSC generation.  However, as we are studying human MSC, it necessitates that studies be done in immune-suppressed mice and evaluation of the impact on immune cells in not possible.  Additionally, teasing out the role of metformin as a preventative measure in mouse models are challenging and would require significant additional time and resources and are beyond the scope of the current work.  As such, these additional studies were not included in this manuscript.